

# Influence of plastic film mulch with biochar application on crop yield, evapotranspiration, and water use efficiency in northern China: A meta-analysis

Erastus Mak-Mensah[1], Peter Bilson Obour[2], Eunice Essel[3], Qi Wang[1] and John K. Ahiakpa[4]

[1] College of Grassland Science, Gansu Agricultural University, Lanzhou, Gansu Province, China
[2] Department of Geography and Resource Development, University of Ghana, Accra, Greater Accra, Ghana
[3] Department of Applied Biology, University for Development Studies, Tamale, Northern region, Ghana
[4] Research Desk Consulting Ltd, Accra, Ghana

Corresponding author
Qi Wang, wangqigsau@gmail.com

## ABSTRACT

**Background**. China is the leading consumer of plastic film worldwide. Plastic film mulched ridge-furrow is one of the most widely adopted agronomic and field management practices in rain-fed agriculture in dry-land areas of China. The efficiency of plastic film mulching as a viable method to decrease evapotranspiration (ET), increase crop yields, and water use efficiency (WUE), has been demonstrated extensively by earlier studies.

**Methods**. A comprehensive evaluation of how co-application of plastic-film mulch and biochar in different agro-environments under varying climatic conditions influence ET, crop yield, WUE, and soil microbial activity were assessed. We performed a meta-analysis using the PRISMA guideline to assess the effect of plastic-film mulched ridge-furrow and biochar on ET, yield, and WUE of wheat (*Triticum aestivum* L.), potato (*Solanum tuberosum* L.), and maize (*Zea mays* L.) in northern China.

**Results**. The use of plastic film increased average yields of wheat (75.7%), potato (20.2%), and maize (12.9%) in Gansu, Ningxia, Shaanxi, and Shanxi provinces, respectively due to the reduction in ET by 12.8% in Gansu, 0.5% in Ningxia, and 4.1% in Shanxi, but increased in Shaanxi by 0.5% compared to no-mulching. These changes may be attributed to the effect of plastic film mulch application which simultaneously increased WUE by 68.5% in Gansu, 23.9% in Ningxia, 16.2% in Shaanxi, and 12.8% in Shanxi, respectively. Compared to flat planting without mulching, in three years, the yield of maize increased with the co-application of plastic film and biochar by 22.86% in the Shanxi and Shaanxi regions.

**Conclusion**. Our analysis revealed co-application of plastic film with biochar is integral for improving soil and water conservation in rain-fed agriculture and as an integrated practice to avert drought while simultaneously mitigating runoff and erosion.

## INTRODUCTION

Poor soil fertility and water scarcity pose a major threat to crop production to meet the food needs of the increasing global population (*Qin, Hu & Oenema, 2015*). Soil water conservation has been identified as an important strategy for enhancing crop productivity in rain-fed agriculture (*Ding et al., 2018*). The amount of soil water and nutrient during different growing seasons have marked impact on crop yields in rain-fed agriculture, especially in semi-arid regions with rapidly changing climate (*Grassini et al., 2010*). Unfortunately, most soils in rain-fed farming areas are nutrient-deficient and susceptible to soil erosion and runoff (*Liu et al., 2009*). Thus, soil as an important natural asset should be properly managed to ensure sustainable agricultural production (*Panpatte & Jhala, 2019*). Appropriate land and water management practices are required to reduce the risk of widespread water resource depletion in dry agricultural areas (*Liu et al., 2014b*). For instance, *Olsovska et al. (2016)* reported drought-induced accelerated leaf diffusion resistance against carbon dioxide ($CO_2$ ($g_m$)) flow resulting in decreased stomatal conductance ($g_s$), leaf mesophyll conductance for $CO_2$, and net $CO_2$ assimilation rate ($A_N$) in wheat. Hence, rain-fed crop production and management practices need to be optimized to provide more resilient options to cope with decreasing precipitation and extreme drought periods in these regions (*Verhulst et al., 2011*).

Soil water conservation by soil mulching has been projected as a feasible approach to overcome water scarcity for crop productivity in rain-fed agricultural areas. Local farmers in the rain-fed agricultural areas of the Loess Plateau of China practice ridge-furrow rainwater harvesting with plastic film mulching to improve yield and water use efficiency of crops (*Eldoma et al., 2016*; *Yu, Jia & Zhao, 2018*; *Zhang et al., 2018*; *Pan et al., 2019*). Mulching offers significant agro-ecological potential (*Erenstein, 2003*) and thus, one of the important agronomic practices to improve moisture retention capacity of soils (*Ye & Liu, 2012*), promotes carbon dioxide ($CO_2$) retention in leaves (*Samui et al., 2020*), soil microbial characteristics, and crop nutrients assimilation (*Chakraborty et al., 2008*). In unproductive soils, plastic film mulching also promotes nutrient use efficiency. For instance, *Mondal et al. (2020)* demonstrated 50% of the recommended dose of nitrogen with no rhizobium resulted in maximum nitrogen use efficiency while under polythene mulch; significant root nodules were recorded for treatments that received 75% of the recommended dose of nitrogen with rhizobium inoculation.

Plastic film mulch reduces evapotranspiration and enhances plant growth (*Qin, Hu & Oenema, 2015*; *Shen et al., 2019*). Plastic mulches usually leave residues in fields they have previously been applied (*Jabran, 2019*). The residual effect of plastic mulching considerably increased yields, and water use efficiencies of *Triticum aestivum* L. and *Zea mays* L. (*Qin, Hu & Oenema, 2015*) while reducing evapotranspiration (ET) (*Fan et al., 2017*). Contrarily, ET increased by 38.1 and 9.3% on plastic film mulched ridge-furrow and flat-planted non-mulched maize fields, respectively (*Gong et al., 2017*). In the first and second seasons with plastic film mulching and flat planting (FP) with no-mulching areas, *Mbah & Nwite (2010)* recorded an increment in yield from 55–78 and 108–142%. In two consecutive growing seasons in China, plastic film mulching with biochar modification increased

the root and shoot biomass and grain yield of maize (*Xiao et al., 2016*). Although plastic film mulching has been the ultimate choice of mulching material in rain-fed areas, to enhance water availability in the soil for plant growth (*Zhang et al., 2017*), it equally poses a challenge of residual plastic film on farmlands which can impede soil structure, plant growth, nutrients and water uptake (*Liu, He & Yan, 2014*). The persistence of residuals in soils from pesticides (*Hüffer et al., 2019*) and fertilizers (*Anyaoha et al., 2018*) pose risks to their continuous use as inputs in agriculture. Consequently, biochar applications with plastic film mulching have been touted as an effective agronomic practice to mitigate the negative effects of residual plastic film mulching under field conditions. However, studies on the co-application of biochar and plastic film mulches in China are limited (*Aller et al., 2018*).

Biochar is a carbon-rich product of the thermo-chemical conversion of organic materials used as soil amendments due to their gradual decomposition rate and influence on nutrient dynamics (*Gao et al., 2019*). The focus of biochar research has advanced from its effects on semi-arid soils to its potential as a soil management material for global agriculture (*Karer et al., 2013*). In arid areas, biochar application improves soil water adsorption capacity, fertility, microbial activity, organic matter content, soil porosity, water retention, soil quality, soil aeration, and nutrients uptake for enhanced crop production (*Yang & Ali, 2019*). Biochar has appreciable carbon sequestration value and may act as a modifier or carbon sink to reduce $CO_2$ emissions from decaying biomass, nutrient leaching, soil bulk density, erosion, or fertilizer needs (*Mohan et al., 2014*; *Kavitha et al., 2018*). The shared impact of plastic film mulching with biochar on ET, crop yield, and WUE as a ridge-furrow rainwater harvesting technology in China are currently less understood (*Nelissen et al., 2012*; *Fischer et al., 2019*). Therefore, understanding how biomass in China and across the world can change under the combined application of plastic film and biochar and processes activated as a result of these changes is key to harnessing their potential for wider use in agriculture (*Antala et al., 2020*).

The effects of plastic film mulched ridge-furrow with biochar on ET, crop yield, and WUE in rain-fed agro-ecological areas in China have been reported in the past with mixed results. We, therefore, hypothesized that the co-application of plastic film with biochar in semi-arid regions is an optimum agronomic practice for minimizing the adverse impact of drought while simultaneously mitigating runoff and erosion. Here, we performed a meta-analysis on relevant literature using the PRISMA guideline (*Moher et al., 2009*) to ascertain the impact of ridge-furrow plastic film mulching with biochar on ET, crop yield, and WUE of maize, wheat, and potato.

## MATERIALS & METHODS

### Data collection

Data from only peer-reviewed publications in English investigating the effects of plastic film mulching and biochar on field crops from 1990–2020 were retrieved from online databases (*ISI Web of Science, Scopus (Elsevier), ScienceDirect, PubMed, JSTOR, and Google Scholar*). Nevertheless, articles from conference proceedings were excluded from this meta-analysis.
In the databases, 'yield', and/or 'plastic film', and/or 'biochar,' and 'mulching' were used as search keywords. Erastus Mak-Mensah and Eunice Essel performed the Search Strategy and independently decided on appropriate publications for the study. Qi Wang intervened and resolved by discussing cases where Erastus Mak-Mensah and Eunice Essel had disagreements on the use of a particular reference in the study. The search produced a total of 556 publications, which were screened based on (1) on-field experimentation containing at least plastic film mulched ridges and no mulch treatments; (2) experimental sites located in rain-fed agriculture areas of China in Gansu, Ningxia, Shaanxi, and Shanxi provinces; (3) colors of the plastic film were black and transparent; (4) the publication included estimates of ET, crop yield, or WUE. Subsequently, due to insufficient and missing data, 535 papers were excluded from this meta-analysis and the final analysis was conducted on 21 studies (papers) based on ET, yields, and WUE after the screening process. The process of screening of publications for the meta-analysis is depicted in a flowchart (Fig. 1); which was adapted from the PRISMA protocol (*Moher et al., 2009*). Farming provinces and locations of field experiments for all the crops in this study are shown in Table 1 and Fig. 2. Data within the selected publications were categorized based on estimated biophysical parameters (Table 2). Variations in ET, yield, and WUE of wheat, maize, and potato under plastic film and no-mulching applications were shown in Table 3 while Table 4 shows the mean, range, and coefficient of variation (CV) of ET, yield, and WUE in different locations and precipitations in northern China. The mean, range, and coefficient of variation (CV) of yield of maize for plastic film mulched ridge-furrow and no-mulching in Shanxi and Shaanxi provinces in China are shown in Table 5.

## Data analysis

Meta-analysis enables the statistical analysis of effect sizes and quantitative evaluation of experimental outcomes reported by other authors. Meta-analysis enhances the statistical capacity available for testing the hypotheses and the reaction variations between treatments in different environments. Unbiased estimation of the underlying true effect size, subject to random variance, can be assumed to be the effect size observed in each sample. The Newcastle Ottawa Scale (NOS), (*Zeng et al., 2015*), was used to assess the importance of the papers involved in this study. High-quality publications (papers) were considered based on $\geq 7$ score. The scores for NOS varied from 6 to 9 (Table 1). More weight is given to data from experiments with more reliable measurements because they have a larger effect on the overall calculation (*Yu et al., 2018*).

We used the construction confidence interval analysis (*Gao et al., 2019*) to correlate the severity of the response ratio between the plastic film mulched ridge-furrow and no-mulching treatments. The effect size was computed as the natural log (ln R) of the response ratio (R) (*Gao et al., 2019*; *Qin, Hu & Oenema, 2015*), which reflects the severity of the effect of plastic film mulch on ET, yield, and WUE in this meta-analysis (*Hedges, Gurevitch & Curtis , 1999*), Eq. (1):

$$R = \theta_t / \theta_c \tag{1}$$
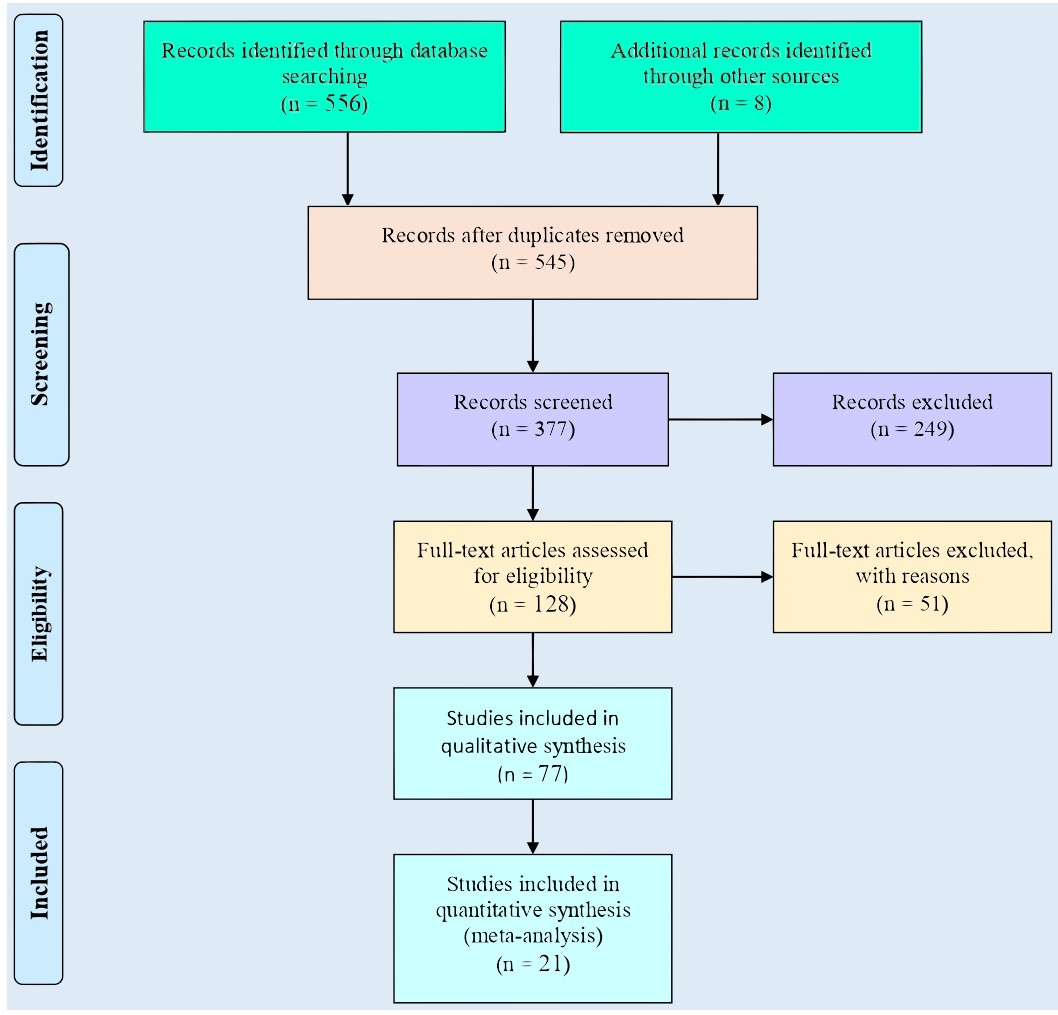

**Figure 1** **Flowchart of literature identification, and screening for use in this study.** Adapted from PRISMA (*Moher et al., 2009*).

$$InR = In(\theta_t/\theta_c) = In\theta_t - In\theta_c \tag{2}$$

where $\theta_t$ and $\theta_c$ equates the mean values of ET, yield, and WUE in plastic film mulched ridge-furrow and no-mulching, respectively. To further authenticate the outcomes from this analysis, the percentage of change (Z) in ET, yield, and WUE were determined according to *Li et al. (2018a)*, *Li et al. (2018b)* as:

$$Z = (R - 1) \times 100\% \tag{3}$$

where a negative value for percentage change shows a decline in the variable with plastic film mulching relative to no-mulching and a positive value for percentage change, indicates an enhancement in the matching variable for plastic film mulching relative to no-mulching. Conversely, the sample sizes of the variables and standard deviation (SD) involved were

| Province | Study areas | Geo-coordinate (N, E, m a.s.l) | Crop | Reference | NOS |
|---|---|---|---|---|---|
| Gansu | Qingyang | 35°42′, 107°20′ | Wheat | *Gao et al. (2014)* | 9 |
| | Tangjiabu, Dingxi | 35°57′, 104°59′, 1970 | | *Li et al. (2004)* | 8 |
| | | 35°33′, 104°35′, 1896.7 | | *Zhao et al. (2012)* | 7 |
| | Dingxi | 35°33′, 104°35′, 1896.7 | | *Zhao et al. (2012)* | 7 |
| | | 35°33′, 104°35′, 1874 | Potato | *Qin et al. (2016)* | 8 |
| | | 36°02′, 104°25′, 2400 | | *Zhao et al. (2014)* | 6 |
| | Zhonglianchuan, Yuzhong | 36°02′, 104°25′, 2400 | | *Liu & Siddique (2015)* | 8 |
| | | 36°2′, 104°25′, 2400 | Maize | *Eldoma et al. (2016)* | 8 |
| | | 36°02′, 104°25′, 2400 | | *Zhou et al. (2009)* | 9 |
| | Gaolan | 36°2′, 103°7′, 1780 | | *Wang et al. (2005)* | 6 |
| | Yuzhong | 35°9′, 104°1′, 1800 | Potato | | 6 |
| Ningxia | Pengyang | 35°51′, 106°48′, 1658 | | *Wu et al. (2017)* | 7 |
| | | 106°45′, 35°79′, 1800 | | *Zhang et al. (2017)* | 8 |
| | | 34°59′, 107°38′, 1220 | | *Lu et al., 2020* | 8 |
| | Changwu | 35°14′, 107°41′, 1206 | Maize | *Zhang et al., 2011* | 8 |
| | | 35°14′, 107°41′, 1200–1206 | | *Lin et al. (2019)* | 6 |
| Shaanxi | | 35°14′, 107°42′, / | | *Qin et al. (2018)* | 6 |
| | | 35°12′, 107°45′, 12000 | Wheat | *He et al. (2016)* | 9 |
| | Heyang | 35°15′, 110°18′, 910 | | *Li et al. (2012)* | 9 |
| | | 35°15′, 110°18′, 910 | Maize | *Han et al. (2013)* | 8 |
| Shanxi | Shouyang | 37°54′, 113°09′, 1273 | | *Gaimei et al. (2017)* | 7 |
| | | 37°45′58″, 113°12′9″, 1202 | | *Gong et al. (2017)* | 7 |

obtained in addition to the means from the articles or computed using the following equation (*Yu et al., 2018*):

$$SD = SE \times \sqrt{n}. \tag{4}$$

For studies which did not report SD; the average coefficient of variation (CV) within each data was computed and then approximated as the unavailable SD using the following equation (*Yu et al., 2018*):

$$SD = CV \times \theta \tag{5}$$

where $\theta$ equates the mean of plastic film mulched ridge-furrow with biochar or no-mulching. The effect sizes of plastic film with biochar and no-mulching for ET, crop yield, and WUE were continuous variables, hence were calculated by random-effects models using the Review Manager software (RevMan; ver. 5.3, Nordic Cochrane Centre, Denmark). The heterogeneity between studies used in this analysis has been measured with $Chi^2$ and $I^2$ statistics (Table 6). The parameters for heterogeneity for the $I^2$ test were as follows: $I^2$ <25% indicates no heterogeneity; moderate heterogeneity is considered to be 25–75%; strong heterogeneity is considered to be $I^2$ >75% (Table 7). Random-effects models were implemented in cases of mild to high heterogeneity, indicated by a $Chi^2$ $p$-value< 0.05

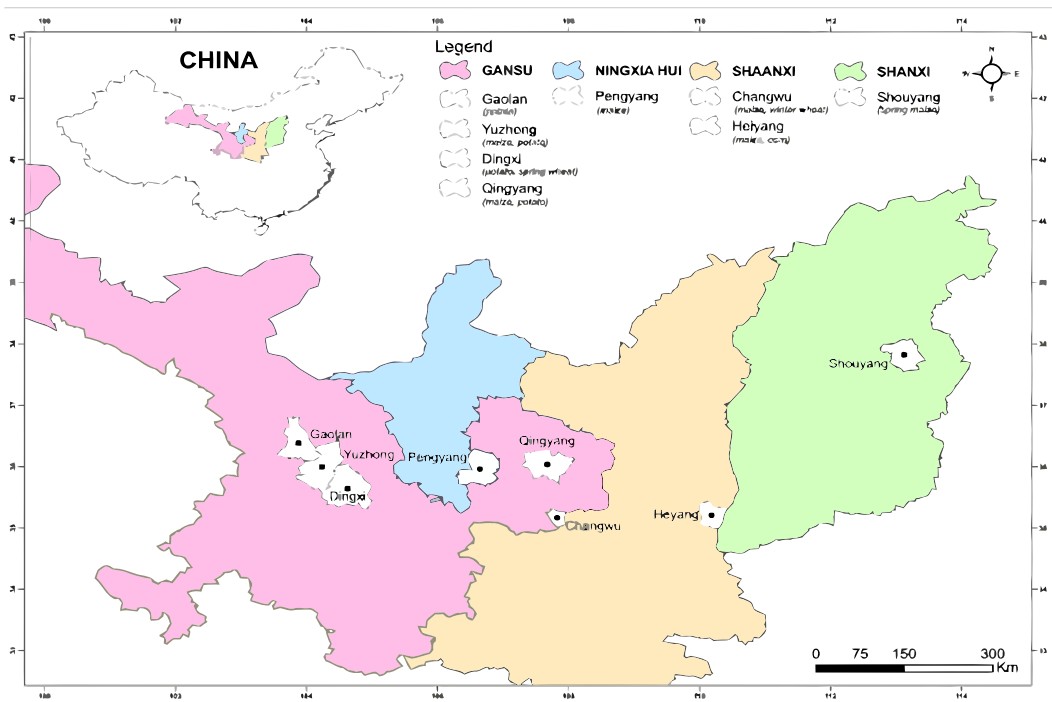

**Figure 2   Experimental locations from the peer-reviewed publications for the meta-analysis** ArcGIS 10.6 software (ESRI, Redlands, California) was used to produce the map.

and $X^2 > 50\%$. The RevMan program weighed the mean differences of the plastic film with biochar and no-mulching groups according to their SE and sample sizes, and their confidence intervals (CI) were computed from their weighted effect sizes. The impact of a treatment was significant if there was no zero in the 95% CIs of the effect size of that treatment. Conversely, the treatment was considered not significant when the 95% CIs includes zero. Similarly, a general linear model in SPSS statistical software (ver. 26.0, SPSS Inc., Chicago, USA) was used to compute the effect of location, crop type, and rainfall on ET, crop yield, and WUE. The frequency distribution of effect sizes (Odds ratio) was computed using Excel 2016 spreadsheet to illustrate the distribution symmetries of the individual studies.

## RESULTS

### Yield response of wheat, maize, and potato in different locations and climate

Considering climate variables (precipitation and air temperature), the meta-analysis indicated that in the growing-seasons, precipitation and air temperature had no significant ($p > 0.05$) effects on maize, wheat, and potato yields in the plastic film mulched ridge-furrow treatment (Fig. 1). The meta-analysis dataset had pH in all the areas of study as slightly alkaline (>7) hence no comparison was made in that regard (Table 4). Therefore, we investigated in three categorized soil types, i.e., light, medium, and heavy, the impacts

Mak-Mensah et al. (2021), *PeerJ*, DOI 10.7717/peerj.10967

**Table 2  Categorization of data within the selected publications.**

| Annual mean precipitation | Annual air temperature | Organic C content | Soil bulk density (0–20 cm) | Soil texture (0–20 cm) | pH | Soil available N | Soil available P | Soil available K |
|---|---|---|---|---|---|---|---|---|
| <400 mm | <9 °C | <9 g/kg | <1.3 g cm$^{-3}$ | Light: sandy and sandy loam soils | Very acidic: pH < 5 | <50 mg kg$^{-1}$ | <20 mg kg$^{-1}$ | <150 mg kg$^{-1}$ |
| >400 mm | >9 °C | >9 g/kg | >1.3 g cm$^{-3}$ | Medium: loamy sand and loam soils | Acidic: pH 5-6 | >50 mg kg$^{-1}$ | >20 mg kg$^{-1}$ | >150 mg kg$^{-1}$ |
| | | | | Heavy: clay loam, silty clay, and clay soils | Neutral: pH 6-7 | | | |
| | | | | | Slightly alkaline: > 7 | | | |

**Notes.**

[a] < 400 (low mean precipitation); > 400 mm (high mean precipitation).

[b] < 9°C (low mean temperature); > 9°C (high mean temperature).

[c] < 9 g/kg (low organic C content); > 9 g/kg (high organic C content).

[d] < 1.3 (low soil bulk density) g cm$^{-3}$; > 1.3 g cm$^{-3}$ (high soil bulk density).

[e] < 50 (low soil available N) mg kg$^{-1}$; > 50 mg kg$^{-1}$ (high soil available N).

[f] < 20 (low soil available P) mg kg$^{-1}$; > 20 mg kg$^{-1}$ (high soil available P).

[g] < 150 (low soil available K) mg kg$^{-1}$; > 150 mg kg$^{-1}$ (high soil available K).

Mak-Mensah et al. (2021), *PeerJ*, DOI 10.7717/peerj.10967

**Table 3** Variations in yield, evapotranspiration (ET), and water use efficiency (WUE) of wheat, maize, and potato under plastic film and no-mulching application

| Treatments | Parameters | Variable | Yield | | | | ET | | | | WUE | | | |
|---|---|---|---|---|---|---|---|---|---|---|---|---|---|---|
| | | | n | Mean | Range | CV | n | Mean | Range | CV | n | Mean | Range | CV |
| Plastic film | Location | Gansu | 10 | 8821.6 | 2162.3–45882 | 151 | 7 | 279 | 215.4–386.5 | 22 | 7 | 33.3 | 0.8–129.95 | 138 |
| | | Ningxia | 2 | 12926 | 12779.3–13072.5 | 1.6 | 2 | 435 | 375.5–494.3 | 19 | 2 | 30.4 | 26.8–34.1 | 17 |
| | | Shaanxi | 7 | 9313.1 | 4931.8–13079.3 | 32.6 | 3 | 367 | 300–409.5 | 16 | 3 | 25.5 | 22–32.1 | 22.2 |
| | | Shanxi | 2 | 11408 | 11290–11526.7 | 1.47 | 2 | 391 | 345.4–435.7 | 16 | 2 | 14.9 | 3.4–26.5 | 110 |
| | Crop type | Maize | 13 | 9813.4 | 2420–13079.3 | 32.8 | 8 | 392 | 300–494.3 | 15 | 7 | 23.9 | 3.4–34.1 | 42.3 |
| | | Wheat | 2 | 3547.1 | 2162.3–4931.8 | 55.2 | 1 | 273 | – | – | 1 | 0.75 | – | – |
| | | Potato | 6 | 11235 | 2359.3–45882 | 152 | 5 | 259 | 215.4–333.7 | 18 | 6 | 38.7 | 6.4–129.95 | 123 |
| | Rainfall | <400 | 8 | 9532.4 | 2359.3–45882 | 156 | 6 | 281 | 215.4–386.5 | 23 | 6 | 38.7 | 6.4–129.96 | 123 |
| | | >400 | 13 | 9678.2 | 2162.3–13079.3 | 35.3 | 8 | 378 | 272.5–494.3 | 19 | 8 | 21 | 0.8–34.1 | 59.2 |
| | Temperature | <9 | 13 | 9776.2 | 2162.3–45882 | 119 | 11 | 328 | 215.4–494.3 | 27 | 11 | 29.4 | 0.75–129.95 | 125 |
| | | >9 | 8 | 9660.8 | 4931.8–13079.3 | 28.7 | 3 | 367 | 300–409.5 | 16 | 3 | 25.5 | 22–32.07 | 22.2 |
| No mulching | Location | Gansu | 10 | 5021.3 | 353–27385.5 | 162 | 7 | 320 | 253.5–461.1 | 26 | 7 | 19.7 | 0.6–79.6 | 144 |
| | | Ningxia | 2 | 10755 | 9978.3–11532 | 10.2 | 2 | 437 | 400–473.99 | 12 | 2 | 24.6 | 24.2–24.9 | 1.93 |
| | | Shaanxi | 7 | 8249.1 | 4650.4–10422.3 | 27.5 | 3 | 365 | 289.7–404 | 18 | 3 | 22 | 19.5–26 | 16.2 |
| | | Shanxi | 2 | 10116 | 9988.3–10243.3 | 1.78 | 2 | 407 | 380.6–433.3 | 9.2 | 2 | 13.2 | 2.7–23.7 | 113 |
| | Crop type | Maize | 13 | 7896.5 | 353–11532 | 44 | 8 | 398 | 289.7–473.99 | 13 | 8 | 17.8 | 0.9–26 | 57 |
| | | Wheat | 2 | 2639.8 | 629.1–4650.4 | 108 | 1 | 273 | – | – | 1 | 0.56 | – | – |
| | | Potato | 6 | 6960.7 | 833–27385.5 | 147 | 5 | 313 | 253.5–461.1 | 29 | 5 | 27.4 | 3.6–79.6 | 113 |
| | Rainfall | <400 | 8 | 5537.7 | 353–27385.5 | 163 | 6 | 328 | 253.5–461.1 | 27 | 6 | 22.9 | 0.9–79.6 | 129 |
| | | >400 | 13 | 8107.4 | 629.1–11532 | 38.7 | 8 | 382 | 273.1–473.99 | 18 | 8 | 17.8 | 0.6–26 | 57.5 |
| | Temperature | <9 | 9 | 7069.7 | 353–27385.5 | 122 | 11 | 357 | 253.5–474 | 24 | 11 | 19.4 | 0.56–79.6 | 117 |
| | | >9 | 8 | 7590.7 | 4650.4 -10422.3 | 28.3 | 3 | 365 | 289.7–404 | 18 | 3 | 22 | 19.5–26.03 | 16.2 |

Mak-Mensah et al. (2021), *PeerJ*, DOI 10.7717/peerj.10967

**Table 4 Mean, range, and coefficient of variation (CV) of yield, evapotranspiration (ET), and water use efficiency (WUE) of wheat, maize, and potato under plastic film mulching and no mulching in different locations and precipitations in northern China**

| Treatments | Parameters | Variables | Yield | | | | ET | | | | WUE | | | |
|---|---|---|---|---|---|---|---|---|---|---|---|---|---|---|
| | | | n | Mean | Range | CV | n | Mean | Range | CV | n | Mean | Range | CV |
| Plastic film | Organic C content | <9 | 8 | 10504 | 2162.3–45882 | 140 | 6 | 334 | 215.4–494.3 | 29 | 6 | 32.9 | 0.75–129.95 | 147 |
| | | >9 | 7 | 11369 | 9260–13079.3 | 13.2 | 4 | 403 | 375.5–435.7 | 6.4 | 4 | 28.8 | 22.5–34.07 | 18.3 |
| | Bulk density | <1.3 | 8 | 11190 | 2162.3–45882 | 129 | 6 | 327 | 230.9–435.7 | 23 | 5 | 47.5 | 6.35–129.95 | 103 |
| | | >1.3 | 9 | 9399.7 | 4255.75–13072.5 | 36.6 | 6 | 379 | 259.2–494.3 | 20 | 6 | 22.5 | 3.36–34.07 | 50.4 |
| | pH | >7 | 11 | 11729 | 2420–45882 | 102 | 7 | 381 | 215.4–494.3 | 23 | 7 | 36.5 | 6.35–129.95 | 115 |
| | Soil texture | Light | 5 | 16667 | 2549.8–45882 | 101 | 4 | 340 | 215.4–435.7 | 27 | 4 | 50.5 | 11.7–129.95 | 106 |
| | | Medium | 5 | 7571.2 | 2162.3–13079.3 | 57.8 | 3 | 254 | 230.9–272.5 | 8.4 | 3 | 23.4 | 0.75–52.85 | 114 |
| | | Heavy | 10 | 7938.6 | 2359.3–13072.5 | 50.1 | 6 | 388 | 300–494.3 | 17 | 6 | 18.8 | 3.36–32.07 | 60.8 |
| | N | <50 | 4 | 7221.3 | 2420–13079.3 | 76.8 | 2 | 301 | 215.4–386.5 | 40 | 2 | 8.98 | 6.35–11.62 | 41.5 |
| | | >50 | 6 | 9935.7 | 2162.3–13072.5 | 41.7 | 5 | 388 | 272.5–494.3 | 20 | 5 | 23.2 | 0.75–34.07 | 57.5 |
| | P | <20 | 8 | 9775.4 | 2420–13079.3 | 40.9 | 4 | 423 | 375.5–494.3 | 13 | 4 | 23.4 | 6.35–34.07 | 50.9 |
| | | >20 | 4 | 7636.7 | 2162.3–12545.3 | 57.5 | 4 | 343 | 272.5–409.5 | 20 | 4 | 19.3 | 0.75–32.07 | 68.4 |
| | K | <150 | 7 | 9788.8 | 4931.8–13079.3 | 31.2 | 4 | 384 | 300–435.7 | 15 | 5 | 20.8 | 0.75–32.07 | 57.3 |
| | | >150 | 4 | 20382 | 9794.5–45882 | 83.7 | 3 | 401 | 333.7–494.3 | 21 | 3 | 63.6 | 26.8–129.95 | 90.5 |
| No mulching | Organic C content | <9 | 8 | 6676.5 | 353–27385.5 | 137 | 6 | 374 | 273.1–473.99 | 23 | 6 | 21.5 | 0.56–79.6 | 140 |
| | | >9 | 7 | 8891 | 5282–10422.3 | 20.6 | 4 | 410 | 400–433 | 3.9 | 4 | 23.5 | 19.5–26.03 | 12.1 |
| | Bulk density | <1.3 | 8 | 7499.1 | 353–27385.5 | 118 | 6 | 332 | 253.5–433.3 | 22 | 5 | 31.1 | 0.85–79.6 | 94.3 |
| | | >1.3 | 9 | 7215.8 | 2184.5–11532 | 49.2 | 6 | 386 | 253.8–473.99 | 19 | 6 | 18.1 | 2.7–26.03 | 51.3 |
| | pH | >7 | 11 | 8213.5 | 353–27385.5 | 91.4 | 7 | 417 | 344.1–473.99 | 10 | 7 | 25.4 | 0.85–79.6 | 102 |
| | Soil texture | Light | 5 | 11622 | 833–27385.5 | 83.1 | 4 | 410 | 344.1–461.1 | 12 | 4 | 33 | 3.6–79.6 | 98.9 |
| | | Medium | 5 | 4872.7 | 629–8848.5 | 66.2 | 3 | 260 | 253.5–273.1 | 4.3 | 3 | 14.3 | 0.56–30.9 | 108 |
| | | Heavy | 10 | 6206.8 | 353–11532 | 65.2 | 6 | 392 | 289.7–473.99 | 15 | 6 | 15.6 | 0.85–26.03 | 70.5 |
| | N | <50 | 4 | 4989.9 | 353–9925.2 | 102 | 2 | 431 | 400–461.05 | 10 | 2 | 2.23 | 0.85–3.6 | 87.4 |
| | | >50 | 6 | 7606 | 629.1–11532 | 53.6 | 5 | 390 | 273.1–473.99 | 19 | 5 | 19 | 0.56–26.03 | 55.8 |
| | P | <20 | 8 | 7569.7 | 353–11532 | 50 | 4 | 427 | 400–473.99 | 8.2 | 4 | 18.4 | 0.85–24.9 | 63.6 |
| | | >20 | 4 | 6172.1 | 629.1–10422.3 | 67.1 | 4 | 342 | 273.1–404 | 21 | 4 | 16.6 | 0.56–26.03 | 66.7 |
| | K | <150 | 7 | 8210.3 | 4650.4–10422.3 | 27.2 | 4 | 382 | 289.67–433.3 | 17 | 5 | 18 | 0.56–26.03 | 56.1 |
| | | >150 | 4 | 13544 | 5282–27385.5 | 70.9 | 3 | 406 | 344.1–473.99 | 16 | 3 | 42.9 | 24.23–79.6 | 74.1 |

**Table 5 Mean, range, and coefficient of variation (CV) of yield of maize for plastic film mulched ridge-furrow and no mulching in Shanxi and Shaanxi provinces in China**

| Treatments | Crop | n | Mean | Range | CV |
|---|---|---|---|---|---|
| Plastic film + biochar mulching | Maize | 3 | 11.913 | 10.43–14.7 | 20.3 |
| No-mulching | Maize | 3 | 9.6967 | 9.11–9.99 | 5.24 |

of ridge-furrow plastic film mulching on maize, wheat, and potato yields (Table 4). In the plastic film mulched ridge-furrow treatment, the mean effect size for the light soil type (1.68 [0.38–2.99]) was significant ($p = 0.01$) as compared to the medium and heavy soil types (Fig. 3). The mean effect size was not significantly ($p > 0.05$) different among the medium and heavy soil types in the plastic film mulched ridge-furrow treatment. Maize yields in Shanxi ranged from 11,290 to 11,527 kg ha$^{-1}$ in the plastic film mulched ridge-furrow treatment and were significantly ($p < 0.05$) higher than for Ningxia which ranged from 12,779 to 13,073 kg ha$^{-1}$ in our meta-analysis dataset (Table 3). The impacts of plastic film mulched ridge-furrow on yield varied with the soil bulk density (Table 4). Plastic film mulched ridge-furrow significantly ($p < 0.05$) improved yield in light soils by 43% compared with flat planting with no-mulching in areas with a soil bulk density of >1.3 g cm$^{-3}$ (Fig. 3). Soil organic carbon (SOC) content of 0–10 cm soil layer in areas of >9 g kg$^{-1}$) in the plastic film mulched ridge-furrow treatments was improved (27.8%) compared with flat planting with no-mulching. With high soil available N (>50 mg kg$^{-1}$), plastic film mulching exerted a greater impact on maize, wheat, and potato yield with high soil available $P$ (>20 mg kg$^{-1}$) and low soil available K (<150 mg kg$^{-1}$).

## ET and water use efficiency of wheat, maize, and potato in different locations

Compared with flat planting without mulching, plastic film mulched ridge-furrow significantly increased WUE (16.1%; $p = 0.01$) in regions with an air temperature >9 °C, but, had no significant impact on ET (0.46%; $p = 0.64$) (Fig. 4). This increase in WUE was significant in regions with heavy soil type and texture (20.68%; $p = 0.01$), soil organic carbon content of > 9 g kg$^{-1}$ (22.2%; $p = 0.03$), and soil available N of > 50 mg kg$^{-1}$ (22%; $p = 0.01$) (Fig. 5). In contrast, plastic film mulched ridge-furrow had no significant effects on ET in heavy soil type (0.99%; $p = 0.96$), soil organic carbon content of > 9 g kg$^{-1}$ (1.67%; $p = 0.91$) and soil available N of > 50 mg kg$^{-1}$ (0.51%; $p = 0.95$) (Fig. 4). The average WUE of maize in Ningxia was significantly increased by 33.9% ($p = 0.01$) with plastic film mulched ridge-furrow higher than 16.2% in Shaanxi compared to flat planting without mulching (Fig. 5). The increase in WUE with plastic film mulched ridge-furrow may be attributed to the increase in yield and decrease in ET, as demonstrated by our analysis.

## Influence of co-application of plastic film mulched ridge-furrow and biochar on yield

In three years, the yield of maize increased significantly with the co-application of plastic film and biochar by 22.86% ($p = 0.05$) compared with flat planting without mulching

**Table 6   Heterogeneity analysis on yield, evapotranspiration (ET), and water use efficiency (WUE) of wheat, maize, and potato under plastic film and no-mulching treatments using random-effects models**

| Items | Parameters | Categories | n | Heterogeneity | | | |
|---|---|---|---|---|---|---|---|
| | | | | df | P | $Chi_2$ | $I_2$ (%) |
| Yield | Location | Gansu | 22 | 9 | 1 | 0.68 | 0 |
| | | Ningxia | 5 | 1 | 0.37 | 0.82 | 0 |
| | | Shaanxi | 27 | 6 | 0.59 | 4.67 | 0 |
| | | Shanxi | 5 | 1 | 0.9 | 0.01 | 0 |
| | Crop type | Maize | 39 | 12 | 0.2 | 15.72 | 24 |
| | | Wheat | 7 | 1 | 0.79 | 0.07 | 0 |
| | | Potato | 14 | 5 | 0.99 | 0.44 | 0 |
| | Rainfall | <400 | 18 | 7 | 1 | 0.55 | 0 |
| | | >400 | 43 | 13 | 0.92 | 6.66 | 0 |
| ET | Location | Gansu | 14 | 6 | 1 | 0.3 | 0 |
| | | Ningxia | 5 | 1 | 0.53 | 0.4 | 0 |
| | | Shaanxi | 10 | 2 | 0.71 | 0.68 | 0 |
| | | Shanxi | 6 | 1 | 0.35 | 0.87 | 0 |
| | Crop type | Maize | 23 | 7 | 0.88 | 3.05 | 0 |
| | | Wheat | 2 | – | – | – | – |
| | | Potato | 10 | 4 | 0.99 | 0.22 | 0 |
| | Rainfall | <400 | 12 | 5 | 1 | 0.29 | 0 |
| | | >400 | 23 | 7 | 0.89 | 2.99 | 0 |
| WUE | Location | Gansu | 14 | 6 | 1 | 0.37 | 0 |
| | | Ningxia | 5 | 1 | 0.19 | 1.71 | 41 |
| | | Shaanxi | 10 | 2 | 0.33 | 2.19 | 9 |
| | | Shanxi | 6 | 1 | 0.5 | 0.46 | 0 |
| | Crop type | Maize | 23 | 7 | 0.69 | 4.79 | 0 |
| | | Wheat | 2 | – | – | – | – |
| | | Potato | 10 | 4 | 0.99 | 0.35 | 0 |
| | Rainfall | <400 | 12 | 5 | 1 | 0.37 | 0 |
| | | >400 | 23 | 7 | 0.68 | 4.87 | 0 |

in the Shanxi and Shaanxi regions. Although, in the plastic film mulched ridge-furrow and biochar co-application treatments, the mean effect size for maize (0.79 [−0.92–2.50]; $p = 0.05$) was not significant as compared to the flat planting without mulching in these regions. Mean crop yields ranged from 10.43 –14.7 (t ha$^{-1}$) (10,430–14,700 kg ha$^{-1}$) with plastic film mulched ridge-furrow and biochar combination treatment as compared to 9.11–9.99 (t ha$^{-1}$) (9,110–9,990 kg ha$^{-1}$) in the flat planting without mulching (Table 5).

## DISCUSSION

In the Loess Plateau, variability in the amount and distribution of seasonal precipitation is a major source of variation in ET, which includes evaporation from the soil surface and crop transpiration (*Lu et al., 2014*). This meta-analysis indicates the yield of wheat, maize, and potato was increased with plastic film mulching compared with flat planting with

**Table 7  Heterogeneity analysis on yield, evapotranspiration (ET), and water use efficiency (WUE) of wheat, maize, and potato under plastic film and no-mulching treatments using random-effects models**

| Items | Parameters | Categories | n | Heterogeneity | | | |
|---|---|---|---|---|---|---|---|
| | | | | df | P | Chi$_2$ | I$_2$ (%) |
| Yield | Organc C content | <9 | 20 | 7 | 1 | 0.45 | 0 |
| | | >9 | 20 | 6 | 1 | 0.28 | 0 |
| | Bulk density | <1.3 | 25 | 7 | 0.97 | 1.78 | 0 |
| | | >1.3 | 25 | 8 | 0.9 | 3.43 | 0 |
| | pH | >7 | 30 | 10 | 0.97 | 3.51 | 0 |
| | Soil texture | Light | 13 | 4 | 0.99 | 0.25 | 0 |
| | | Medium | 10 | 4 | 0.99 | 0.32 | 0 |
| | | Heavy | 34 | 9 | 0.93 | 3.68 | 0 |
| | N | <50 | 9 | 3 | 0.99 | 0.09 | 0 |
| | | >50 | 16 | 5 | 0.96 | 1.04 | 0 |
| | P | <20 | 22 | 7 | 0.87 | 3.2 | 0 |
| | | >20 | 12 | 3 | 0.78 | 1.11 | 0 |
| | K | <150 | 23 | 6 | 0.76 | 3.4 | 0 |
| | | >150 | 9 | 3 | 0.8 | 1.01 | 0 |
| ET | Organc C content | <9 | 13 | 5 | 0.99 | 0.48 | 0 |
| | | >9 | 13 | 3 | 0.69 | 1.48 | 0 |
| | Bulk density | <1.3 | 14 | 5 | 1 | 0.33 | 0 |
| | | >1.3 | 17 | 5 | 0.74 | 2.74 | 0 |
| | pH | >7 | 18 | 6 | 0.98 | 1.14 | 0 |
| | Soil texture | Light | 10 | 3 | 0.92 | 0.5 | 0 |
| | | Medium | 6 | 2 | 0.99 | 0.02 | 0 |
| | | Heavy | 17 | 5 | 0.78 | 2.46 | 0 |
| | N | <50 | 4 | 1 | 0.99 | 0 | 0 |
| | | >50 | 14 | 4 | 0.81 | 1.57 | 0 |
| | P | <20 | 10 | 3 | 0.89 | 0.65 | 0 |
| | | >20 | 12 | 3 | 0.87 | 0.72 | 0 |
| | K | <150 | 13 | 3 | 0.87 | 0.72 | 0 |
| | | >150 | 7 | 2 | 0.8 | 0.45 | 0 |
| WUE | Organc C content | <9 | 13 | 5 | 1 | 0.25 | 0 |
| | | >9 | 13 | 3 | 0.33 | 3.41 | 12 |
| | Bulk density | <1.3 | 12 | 4 | 1 | 0.17 | 0 |
| | | >1.3 | 17 | 5 | 0.8 | 2.38 | 0 |
| | pH | >7 | 18 | 6 | 0.91 | 2.09 | 0 |
| | Soil texture | Light | 10 | 3 | 0.48 | 2.47 | 0 |
| | | Medium | 6 | 2 | 0.95 | 0.11 | 0 |
| | | Heavy | 17 | 5 | 0.95 | 2.41 | 0 |
| | N | <50 | 4 | 1 | 0.86 | 0 | 0 |
| | | >50 | 14 | 4 | 0.62 | 2.66 | 0 |
| | P | <20 | 10 | 3 | 0.49 | 2.43 | 0 |
| | | >20 | 12 | 3 | 0.52 | 2.28 | 0 |
| | K | <150 | 15 | 4 | 0.62 | 2.61 | 0 |
| | | >150 | 7 | 2 | 0.42 | 1.75 | 0 |

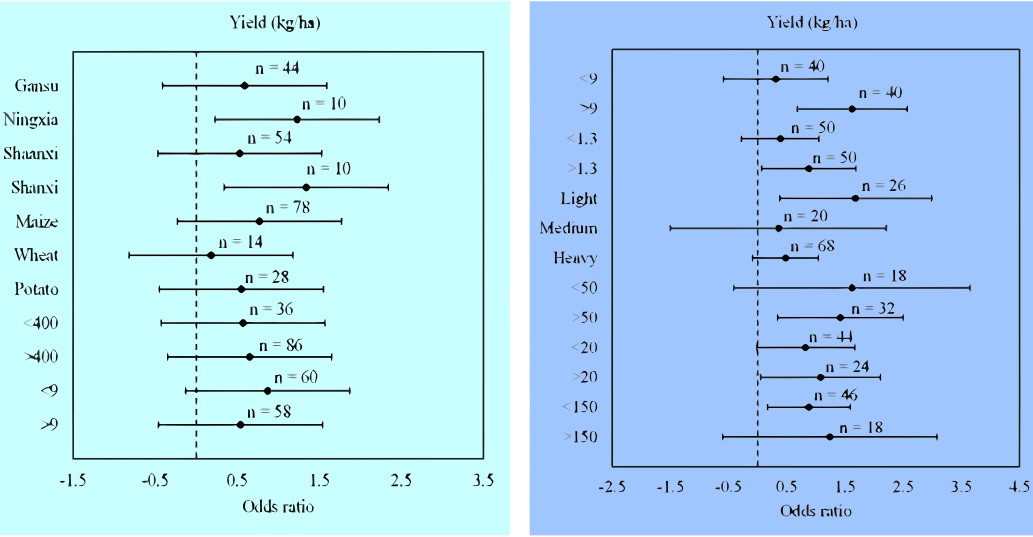

**Figure 3** **(A) Odds ratios of crop yields in different locations and climate. (B) Odds ratios of yield in different soil properties.** The error bars signify 95% confidence intervals, and the values above the bars indicate the number of observations (n).

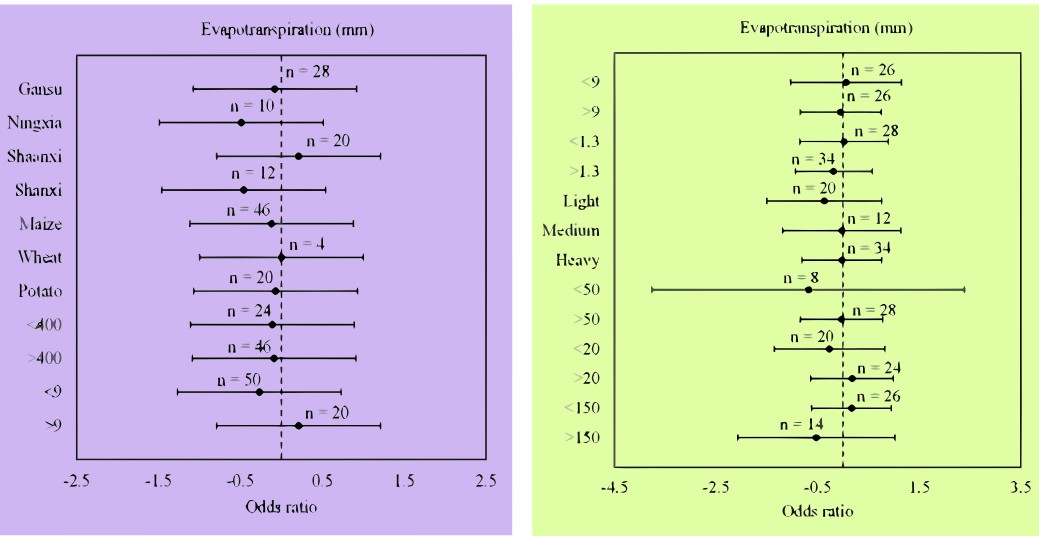

**Figure 4** **(A) Odds ratios of evapotranspiration (ET) in different locations and climate. (B) Odds ratios of evapotranspiration (ET) in different soil properties.** The error bars signify 95% confidence intervals, and the values above the bars indicate the number of observations (n)

no-mulching in Gansu, Ningxia, Shaanxi, and Shanxi provinces. This may be ascribed to increased WUE and decreased in ET in the treatment fields. This is consistent with *Mbah & Nwite (2010)*, who reported plastic film mulch boosts maize yield (55–78%) in the first and second seasons (108–142%) of maize production. *Ding et al. (2019)* found that with plastic film mulching, soil hydrothermal conditions improved and substantially accelerated

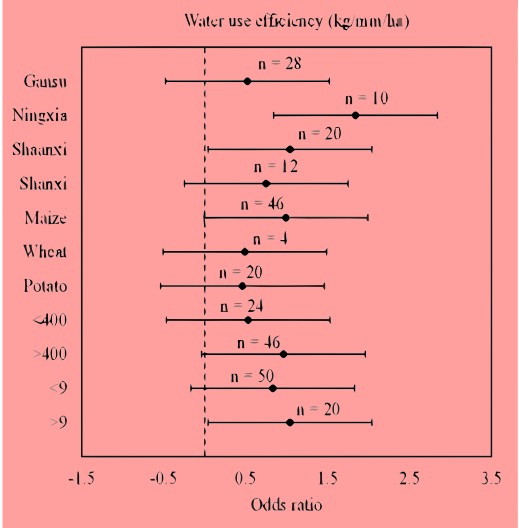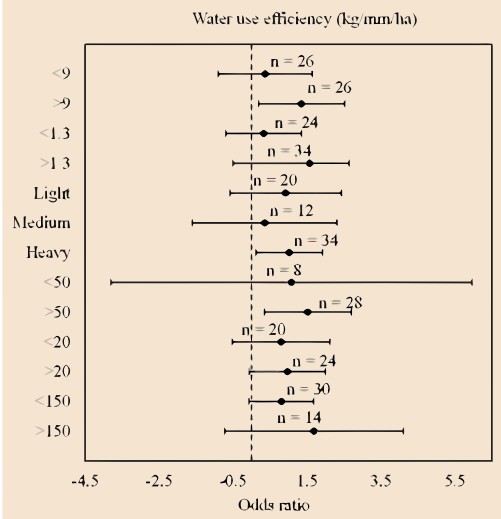

**Figure 5** (A) The odds ratios of water use efficiency (WUE) for plastic film relative to no mulching in different locations and climate. (B) The odds ratios of water use efficiency (WUE) for plastic film relative to no mulching in different soil properties. The error bars show the 95% confidence intervals, and the values above the bars indicate the number of observations (n).

the emergence of wheat leaves and tiller growth, resulting in increased spike number and grain yield. Again, transpiration (*Zhou et al., 2009*), and soil evaporation (*Zribi et al., 2015*) decreased with the application of plastic film mulch hence maize yield was improved. Thus, plastic film mulching significantly improves crop production and increases resource use efficiency, as a potential soil amendment for sustainable dryland farming (*Ding et al., 2019*).

Several studies have subsequently shown that plastic film mulching enhances yield and WUE in different crop fields (*Anikwe et al., 2007*). In this study, plastic film mulching significantly ($p = 0.01$) increased WUE and decreased ET ($p > 0.05$) in the low and high areas of rainfall in Gansu, Ningxia, Shaanxi, and Shanxi provinces. In these areas, the decrease in ET improves the volume of soil water that enhances crop emergence and maturity. This finding is consistent with a research by *Liu et al. (2014a)*; *Liu et al. (2014b)*, which asserted full-year double ridge–furrow plastic film mulching could increase grain yields of maize (110 kg N ha$^{-1}$) and conserve soil water during periods of drought. Simulation of soil water and heat flow in ridge cultivation with plastic film mulching on the Chinese Loess Plateau decreased ET where plastic film mulching was less efficient practice for increasing WUE in dryland agriculture (*Zhao et al., 2018*). Plastic film mulching can provide conducive surroundings for attaining high potato yield (*Wang et al., 2019*) and facilitating maize grain filling hence maximizing yield (*Liu et al., 2016*). Consequently, plastic film mulched ridge-furrow approach may serve as a promising agronomic method in arid and semiarid regions to increase potato yield (*Qin et al., 2016*).

The ridge furrow (RF) rainfall harvesting planting with N:P fertilizer rate (300:150 kg ha$^{-1}$) significantly increased ($p < 0.05$) the mean WUE over 2 years by 53% compared with
the traditional flat planting (*Li et al., 2018a*). Conversely, *Zhang et al. (2019)* in a report suggested 50 cm mulched ridge:10 cm bare furrow ridge-plastic film furrow mulching (RFM) system was more effective in increasing maize growth compared to conventional flat planting. This increased maize grain yield and WUE from 43.1% to 59.2% and from 38.5% to 57.4%, respectively. Concurrently, yield, and WUE in a study by *Fan et al. (2019)* revealed improved grain yield of 20.0% and 3.45 kg ha$^{-1}$ mm$^{-1}$ with plastic film mulched ridge-furrow, respectively. Furthermore, *Dang et al. (2016)* in 2014 discovered plastic film-mulched ridge-furrow (RF) used 17.9% less water and 33.1% more WUE than flat planting (FP) with no-mulching. In 2015, RF showed 56.2% higher yield, 15.0% lower water use (ET), and 63.4% higher WUE than FP, respectively. *Zhao et al. (2012)* in 2009 and 2010 also reported yields from plastic film mulched fields which increased from 33.9–92.5% and 62.9–77.8%, respectively, relative to FP, and corresponding WUEs increased from 41.4–112.6% and 45.9–70.6%. Compared to traditional flat planting, the average four-year maize yield increased from 1497.1 kg ha$^{-1}$ to 2937.3 kg ha$^{-1}$ using the ridge and furrow farming method, and the WUE increased from 2.3 kg ha$^{-1}$ mm$^{-1}$ to 5.1 kg ha$^{-1}$ mm$^{-1}$ (*Ren et al., 2016b*). Approximately, in a three-year study, *Ren et al. (2016a)* revealed WUE and yield of winter wheat was significantly higher in a 60 cm ridge with 60 cm furrow width than in the conventional flat planting without ridging by 2.39 kg mm$^{-1}$ ha$^{-1}$, and 405.1 kg ha$^{-1}$ ($p < 0.05$). However, with increases in mulch length, both tuber yield and WUE decreased, indicating plastic film mulch requires early removal (*Wang et al., 2009*). The biodegradable mulch from our analysis improved by 64.5–73.1%, WUE in maize, wheat, and potato compared to FP (*Deng et al., 2019*). In addition, *Xiaoli et al. (2013)* in a three-year field experiment integrating various furrow-applied mulches in maize production under a plastic film mulched ridge and furrow rainwater harvesting (PRFRH) in China's Loess Plateau semi-arid lands revealed a decrease in plastic film with a thickness of 0.08 mm use. This indicates soil evaporation losses may be minimized by mulching and emphasizes the potential to increase crop sustainability via integrated PRFRH systems in semi-arid areas.

*Xiang et al. (2017)* in a meta-analysis revealed biochar modifications increased root biomass by 32%, root diameter by 9.9%, root volume by 29%, root tips by 17%, root length by 52%, and surface area by 39%. Plant roots play key roles in plant maturity (*Yu et al., 2019*). By altering the growth of roots and rhizosphere microbial activities, biochar may accelerate plant growth and nutrient uptake (*Lehmann & Joseph, 2012*). *Joseph et al. (2010)* found plant roots or root hairs enter soil macro-pores filled with water or attach to the surface of biochar, triggering assorted reactions to facilitate absorption of nutrients. Furthermore, the use of biochar by *Mensah & Frimpong (2018)* in maize production on acidic soils in Ghana resulted in a substantial increase ($p < 0.01$) in leaf number, plant height, and stem girth. *Agegnehu et al. (2016)* established significant correlations in maize grain yield with total biomass, leaf chlorophyll, N and P foliar content, soil organic matter, and soil water content as direct effects of biochar application compared to control. *Liu et al. (2014a)* in an experiment obtained the highest yield of sweet potato (53.77%; $p < 0.05$), which was higher compared to no biochar treatment (control). *Liang et al. (2014)*, following biochar application, obtained 10% higher grain yield in winter wheat and summer maize

than control (no biochar). Again, *Liang et al. (2014)* reported an increase in soil pH with increasing biochar application rates.

According to *Xiao et al. (2016)*, 20 and 30 t ha$^{-1}$ biochar treatments increased wheat yields by 9 and 13% in 2012 and 11 and 14% in 2013 compared to no biochar treatments, respectively. Wheat grain yield remarkably improved by 6 and 9% in 2012 and 2013 with plastic film mulched ridge-furrow with 20 t ha$^{-1}$ biochar treatments compared to plastic film mulched ridge-furrow without biochar treatments (*Xiao et al., 2016*). In addition, *Jeffery et al. (2011)* in a meta-analysis indicated biochar-treated soils increased crop productivity averagely at 10% (−28% to 39%) compared with plots without mulching. Residual impact of biochar on soil fertility largely accounted for an increase in crop yield under co-application of plastic film mulched ridge-furrow with biochar treatment (*Rehman & Razzaq, 2017*).

## CONCLUSIONS

In rain-fed agricultural regions with minimal rainfall in cropping seasons, ridge-furrow mulching with plastic film results in improved crop yields and WUE. The co-application of plastic film mulched ridge-furrow with biochar may potentially mitigate the adverse effects of plastic film application including greenhouse gas emissions, and plastic film residue buildup in soils. Our analysis indicates WUE and yield of maize, wheat, and potato in Gansu, Ningxia, Shaanxi, and Shanxi provinces were significantly influenced by the plastic film mulch application compared to control (no-mulching) ($p < 0.05$). Plastic film mulched ridge-furrow approach of farming had a significant ($p = 0.01$) impact on light soil type compared to the medium and heavy soil types. ET was significantly decreased as compared with FP during the planting seasons. The combined application of plastic film mulch with biochar in these regions improved yield by 22.86% compared with FP. This may be an ideal agronomic practice that may be employed by smallholder farmers in crop production for optimum yield. The practice may equally serve as a potential soil and water-saving practice in rain-fed agriculture especially in areas with changing climate to minimize the effect of drought while mitigating runoff and erosion. A future study on plastic film mulched ridge-furrow rainwater harvesting system with biochar may assess and provide detailed information on the combined effect of biochar with plastic film on soil physico-chemical properties under field conditions.

### Funding

This research was funded by the National Natural Science Foundation of China (42061050 and 41661059). The funders had no role in study design, data collection and analysis, decision to publish, or preparation of the manuscript.

### Grant Disclosures

The following grant information was disclosed by the authors:
National Natural Science Foundation of China: 42061050, 41661059.

## Competing Interests

John Kojo Ahiakpa is employed by Research Desk Consulting Ltd.

## Author Contributions

- Erastus Mak-Mensah conceived and designed the experiments, performed the study, analyzed the data, prepared figures and/or tables, authored or reviewed drafts of the paper, and approved the final draft.
- Peter Bilson Obour analyzed the data, authored or reviewed drafts of the paper, and approved the final draft.
- Eunice Essel conceived and co-designed the study, prepared figures and/or tables, and approved the final draft.
- Qi Wang co-conceived and co-designed the study, prepared figures and/or tables, and approved the final draft.
- John K. Ahiakpa analyzed the data, prepared figures and/or tables, authored or reviewed drafts of the paper, and approved the final draft.

## Data Availability

Raw measurements are available in the Supplementary Files.

## Supplemental Information

Supplemental information for this article can be found online at http://dx.doi.org/10.7717/peerj.10967#supplemental-information.

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
