# Peer review of "Influence of plastic film mulch with biochar application on crop yield, evapotranspiration, and water use efficiency in northern China: A meta-analysis"

_PeerJ, doi:10.7717/peerj.10967_

## Round 0.1 · original submission · Major Revisions

I have received reports from two reviewers who have recommended that your manuscript requires major revisions before further considered for publication in PeerJ. I have accepted their advice and invite you to revise your manuscript in the light of reviewer suggestions. I look forward to your revised manuscript.

Reviewer 1 ·

Basic reporting

Manuscript is interesting, but needs a lot of clarifications. Title seems to be experimental manuscript... How you decide about use just only this References. Yes, 2 people search and one decide - l. 109 - 112. Why the criterium about crop yield and WUE, ET is last...

Current publications are missing. I recommend adding: doi.org/10.3390/plants9010043; DOI: 10.1016/j.scitotenv.2018.12.047; DOI: 10.3389/fpls.2016.01111; DOI: 10.3390/su12176779; DOI: 10.3390/agronomy10101513; 10.1016/j.resconrec.2018.04.022

Add more information on the molecular and physiological flexibility of plants (potato, wheat and maize) under biochar application. The discussion section needs to be revised. Arguments require a clearer and more accurate presentation. The understanding of physiological mechanisms is limited because it is limited to works that have a specific view and deliberately ignore alternatives, and do not represent a balanced view of the evidence.

Experimental design

Meta-analysis have big variance...How you measure (calculate) statistical importance of studies? (l. 127)

Do you used cross searching in databases? More connections together? For example: plastic mulch AND biochar AND WUE? More cleary - l. 108

l. 112 - 556 publications on WOS and Scopus or from all sources... Google scholar counted textbooks etc... Needs clarification and using only peer-reviewed papers!

Validity of the findings

It is very hard to check validity - because in Methods is about 556 publications sources, but Table 1 - just only 21 References. We needs to see whole list of references and their impact.

Findings: only indication... it not could be statistically significant..

Additional comments

In my opinion, needs this meta-analysis study more clarifications, cleary results with valuable and actual worldwide (now is focused to the local sources) disscusion (missing mainly in case of physiological parameters WUE, ET, now is very general not pointed).

·

Basic reporting

Required more information and review of previous studies. Also, need to check the consistency of each sentence.

Experimental design

No explanation about experiment design. Further methodology discussions are needed.

Validity of the findings

The result should discuss specific aspects and data needed in conclusions.

Additional comments

Overall comments:
The manuscript describes the film mulching effects on ET, crop productivity in China which have a significant impact on water-saving agriculture. This paper is relevant and within the scope of this journal. However, the structure of the manuscript is well while I found several grammatical mistakes in sentences where a few terms are repeated.
Furthermore, the abstract need to be revised by considering the importance of the study, appropriate objectives, and specific outcomes. The introduction is not much informative and has been found in many misleading sentences which should be discussed clearly. Therefore, the manuscript needs to significant revision and tremendous efforts to be publishing PeerJ journal.
I made several major and minor comments marked by yellow highlighted colors in the attached pdf file. Thank you.

---

## Round 0.2 · accepted · Accept

Following suggestions from the reviewer(s) and after careful assessment of your revised manuscript, I am pleased to inform you I have accepted your manuscript for publication in PeerJ.

Reviewer 1 ·

Basic reporting

Dear Authors,
thank you for your deep revision. Now is meta-analysis manuscript sufficient and clearly readable.

Experimental design

In revised version is clearly described whole meta-analysis design.

Validity of the findings

Completely revised Discussion and clearly Results.

Additional comments

Thank you for improving your manuscript.